# Molecular Characterization and Taxonomic Assignment of Three Phage Isolates from a Collection Infecting *Pseudomonas* *syringae* pv. *actinidiae* and *P.* *syringae* pv. *phaseolicola* from Northern Italy

**DOI:** 10.3390/v13102083

**Published:** 2021-10-15

**Authors:** Gabriele Martino, Dominique Holtappels, Marta Vallino, Marco Chiapello, Massimo Turina, Rob Lavigne, Jeroen Wagemans, Marina Ciuffo

**Affiliations:** 1Institute for Sustainable Plant Protection, National Research Council of Italy, I-10135 Torino, Italy; gabriele.martino@ipsp.cnr.it (G.M.); marta.vallino@ipsp.cnr.it (M.V.); marco.chiapello@ipsp.cnr.it (M.C.); massimo.turina@ipsp.cnr.it (M.T.); 2Laboratory of Gene Technology, Department of Biosystems, Katholieke Universiteit Leuven, 3001 Leuven, Belgium; dominique.holtappels@kuleuven.be (D.H.); rob.lavigne@kuleuven.be (R.L.); jeroen.wagemans@kuleuven.be (J.W.)

**Keywords:** Psa, Pph, phage, biocontrol, bean, kiwifruit, *Pseudomonas syringae*, kiwifruit canker, bean halo blight

## Abstract

Bacterial kiwifruit vine disease (*Pseudomonas* *syringae* pv. *actinidiae*, Psa) and halo blight of bean (*P. syringae* pv. *phaseolicola*, Pph) are routinely treated with copper, leading to environmental pollution and bacterial copper resistance. An alternative sustainable control method could be based on bacteriophages, as phage biocontrol offers high specificity and does not result in the spread of toxic residues into the environment or the food chain. In this research, specific phages suitable for phage-based biocontrol strategies effective against Psa and Pph were isolated and characterized. In total, sixteen lytic Pph phage isolates and seven lytic Psa phage isolates were isolated from soil in Piedmont and Veneto in northern Italy. Genome characterization of fifteen selected phages revealed that the isolated Pph phages were highly similar and could be considered as isolates of a novel species, whereas the isolated Psa phages grouped into four distinct clades, two of which represent putative novel species. No lysogeny-, virulence- or toxin-related genes were found in four phages, making them suitable for potential biocontrol purposes. A partial biological characterization including a host range analysis was performed on a representative subset of these isolates. This analysis was a prerequisite to assess their efficacy in greenhouse and in field trials, using different delivery strategies.

## 1. Introduction

The *Pseudomonas syringae* pathovar group is considered one of the ten most important phytobacteria, having a high scientific and economic importance [1]. Within this species complex, *P. syringae* pv. *actinidiae* (Psa) is one of the most prevalent bacterial agents causing diseases in kiwifruit orchards. Psa can infect both *Actinidia chinensis* var. deliciosa and *A. chinensis* var. chinensis and is widely spread in Italy and New Zealand, the two major production areas worldwide [2]. Psa penetrates the plant through wounds caused by frost, wind and rain, and natural openings such as stomata and hydathodes, causing primary external infections. In this first phase, it causes interveinal leaf spots. After an asymptomatic phase in which the bacteria are dormant, at the end of winter and in the beginning of spring, it multiplies again and migrates with a systemic infection of the xylem resulting in secondary internal infections. This secondary infection is characterized by the appearance of cankers in trunks and vines [2]. In spring and summer, the bacteria actively multiply and invade the vine xylem, which they use as a highway to spread throughout the plant. The symptoms are particularly visible in spring and autumn because of its mild temperature and high humidity conditions which are ideal for bacterial pathogenicity [2,3].

*P. syringae* pv. *phaseolicola* (Pph), by contrast, which is also known as *P. savastanoi* pv. *phaseolicola*, causes bean halo blight, an economically relevant seed-borne disease in beans. It causes water-soaked lesions surrounded by halos in leaves, stems and pods in the common bean, *Phaseolus vulgaris*, and different species and cultivar of beans worldwide [4,5]. The bacteria can also infect several weeds, providing a reservoir from which a new infection can emerge [6].

For both diseases, the main defense strategy used in Europe is an integrated pest management (IPM) approach based on good agricultural practices, resistant crops and copper treatments. However, efficacy of IPM can be hindered by the emergence of new bacterial strains [4], and varieties resistant to the most prevalent Pph race 6 are not available [7]. Moreover, copper treatments are only preventive and were proven to be harmful for kiwifruits and for the environment, and their intensive use is responsible for the selection of copper resistant strains [8,9].

Therefore, bacteriophage-based biocontrol should be studied as a sustainable alternative. Phages are bacterial viruses and are advantageous in that they are very specific and do not leave harmful residues on the crop that would impair human and environmental health [9]. In recent years, the research on phages to tackle phytopathogens is increasing and has been recently reviewed [10]. Regarding the *P. syringae* species complex, phages have been characterized against *P*. *syringae* pathovars *porri* [11], *aesculi* [12] and *morsprunorum* [13,14]. Several studies have described phages that could potentially be used to control Psa [15,16,17,18,19,20,21,22] and Pph [23,24,25].

This study reports the isolation and characterization of two Psa phages, with isolates psageA1 and psageB1 as representatives of two newly defined species, and a group of Pph phages, with pphageB1 as representative of a third new phage species. Some of these phages have been selected as potential biocontrol candidates for a safe and efficient prevention or treatment of kiwifruit canker and bean halo blight, respectively.

## 2. Materials and Methods

### 2.1. Bacterial Isolation

The Psa collection was composed of 22 Psa strains isolated from infected *Actinidia* spp. in the Piedmont and Veneto regions of northern Italy. Seven Psa strains were collected from symptomatic leaves of *Actinidia deliciosa* and bark tissues sampled in the field from the 2018 growing season in Cuneo province (Piedmont). The Psa collection was further expanded with sixteen additional strains pathogenic for *Actinidia deliciosa* and *Actinidia chinensis* from the Phytosanitary Inspection Services of Piedmont and Veneto regions (Table 1). For Pph, ten strains were gathered from infected bean plants from Piedmont and China between 2015 and 2018. Additionally, two isolates of Pph were received from the Phytosanitary Inspection Service of Piedmont (Table 1).

Included in the biological assays were some *P. syringae* strains that are non-pathogenic to kiwifruit and bean. One strain of *P. syringae* pv. *syringae* causing disease in apricot, was kindly donated by the University of Milan, and eight *P. syringae* non-pathogenic strains, whose lack of pathogenicity is caused by the absence of a functioning Type 3 Secretion System, were provided by the French National Research Institute for Agriculture, Food and the Environment (INRAe) [26,27].

Bacterial colonies were identified or confirmed as Psa or Pph pathovars using specific PCR primers as previously described [28,29,30]. The amplified fragments of Psa and Pph were cloned in pGEM-Teasy (Promega), transformed in competent Escherichia coli DH5α cells, and grown in Lysogeny Broth (LB) at 37 °C. The sequences of the amplified fragments were analyzed through Sanger sequencing to confirm species and pathovars (BioFab Laboratories, Rome, Italy).

Strains were maintained in glycerol at −80 °C (15% glycerol). Bacteria were routinely cultured in solid King’s Broth plates (1.5% of agarose) or in low-salt LB medium (LB_LS_) (0.5 g NaCl). Solid LB_LS_ (1.5% agar) and soft LB_LS_ (0.6% agar) were used to grow bacterial strains and perform plaque assays.

### 2.2. Sample Preparation, Phage Isolation, Amplification and Purification

For bacteriophage isolation, plants, soil and wastewater samples were collected near Psa- and Pph-infected plants with typical disease symptoms in Veneto and Piedmont (Table 2). Thirteen additional phages isolated from cherry bacterial canker (caused by *P. syringae* pv. *morsprunorum* race 1, *P. syringae* pv. *morsprunorum* race 2, and *P. syringae* pv. *syringae*) were added to our host range analysis [13]. Phage buffer (10 mM Tris pH 7.5, 10 mM MgSO_4_, 150 mM NaCl) was used for storage, enumeration, stability assays and dilution of phages. Psa strains K7#1 and K7#8, and Pph strains Cuneo #6_17 and Cuneo #6_18, were used as bacterial host strains. The symptomatic leaf and bark samples were crushed with physiological solution (0.8% NaCl *v*/*w*) in stomacher bags and the filtrates were stored at 4 °C before plaque assays.

For each soil sample, a 5 g aliquot was taken and suspended in 20 mL of filtered and deionized water. These samples were gently mixed at 4 °C overnight to release the phages from the soil particles before centrifugation (4000 rpm; 20 min). The supernatant was filtered with 0.45 µm pore size membrane filters (Millipore, Sigma-Aldrich, Gillingham, UK). Next, plaque assays were performed to test the filtrates for the presence of phages using the soft-agar overlay method [31]. Plates were incubated at 26 °C for one night to allow the formation of plaques. The plaques detected with this method were propagated in three successive single plaque isolations to obtain pure phage isolates.

The amplification of the phages was performed by growing host bacteria to an optical density at 600 nm of 0.3 (approximately 10^9^ CFU/mL). Then, 0.5 mL of phage buffer containing phages picked with a sterile toothpick from a plaque were added. The broth was left shaking at 180 rpm and 26 °C overnight. Obtained crude lysates were filtered through 0.22 µm membrane filters (Millipore, Sigma-Aldrich, Gillingham, UK). To concentrate the phages, the crude lysate was purified through virion precipitation with polyethylene glycol (PEG) 8000 (15%). After the addition of PEG to 35 mL of crude lysate, the mixtures were left slowly shaking at 4 °C overnight. The solution was then centrifuged at 4000 rpm (Sorvall GSA rotor, ThermoFisher, Waltham, MA, USA) for 1 h, and the resulting pellet was dissolved in 3 mL of phage buffer and titrated. The purified samples were kept with a concentration of at least 10^8^ PFU/mL. For long-term storage at −80 °C, 700 µL of the amplified phage solution was mixed with 300 µL of 50% glycerol.

### 2.3. Transmission Electron Microscopy

A drop of 10 µL of PEG-purified phages was deposited on carbon and formvar-coated 400 mesh grids (Gilder, Grantham Lincolnshire, England) and left to adsorb for 3 min. Grids were rinsed several times with water and negatively stained with aqueous 0.5% *w*/*v* uranyl acetate; excess solution was removed with filter paper. Observations and photographs were made using a Philips CM 10 transmission electron microscope (Eindhoven, The Netherlands), operating at 60 kV. Micrograph films were developed and digitally acquired at high resolution with a D800 Nikon camera; images were trimmed and adjusted for brightness and contrast using the GIMP 2 software (Ver.2.10.24). The pphageB1, psageA1, and psageB1 phage capsid diameter and tails length were measured using ImageJ (Ver. 1.53e) [32].

### 2.4. Host Range Assay

Bacteriophages were originally isolated using Psa strain K7#8 and Pph strain Cuneo #6_18. The host range of each selected phage was tested through an efficiency of plating (EOP) analysis [33]. Serially diluted phage lysates were applied (5 µL drops) to soft agar containing one of the bacteria strains of our collection (4 mL of melted 0.6% LBLS soft-agar and 250 µL of an overnight culture of Psa or Pph). For each bacteria phage combination three different dilutions were spotted from the same phage crude lysate.

The plates were incubated at 26 °C for 24 h and then the plaques were counted to obtain phage titration. For each phage, all the titration values were compared quantitatively to the titration value of the same phage infecting the strain used for its isolation.

### 2.5. Influence of pH, Temperature, and UV Radiation to Phage Viability

Phages pphageB1, psageA1, psageB1, and psageK4 were exposed to different abiotic stresses (pH, temperature and UV-C radiation) and then titrated using the reference bacterial strains described above. All these tests were performed on crude lysates after amplification in liquid bacterial cultures. For testing resistance to high temperature, phages were exposed to 26 °C, 37 °C and 55 °C for 1 h and then titrated. For testing resistance to UV-C radiation, phages were exposed to radiation from a UV-C lamp (253.7 nm with an ultraviolet output of 4.9 W at a 48 cm distance). In the pH stability assay for each phage, 100 µL of crude lysate were added to 900 µL phage buffer adjusted with NaOH or HCl to a pH of 2, 4, 6, 8 or 10 and left at room temperature for 18 h before titration. The titers of viable phages were compared with the titers of phages left in phage buffer, at pH 7.5. The graphic visualization of the stability test was obtained using the ggplot2 R package [34] and the statistical analysis was performed with the agricolae R package [35].

### 2.6. Phage Genome Sequencing, Assembly, and Annotation

Phage DNA was extracted using a chloroform-phenol extraction protocol after treatment with DNAse I and RNAse A to remove bacterial genetic material and a proteinase K treatment to break the virus capsids and release phage genomic DNA [36].

The phage’s genomic DNA was sequenced in-house at KU Leuven (Laboratory of Gene Technology) in Belgium. Illumina sequencing libraries for each sample were created using the Nextera Flex DNA Library Kit. The quality of the libraries was assured using an Agilent Bioanalyzer 2100. The libraries were then sequenced with a MiniSeq Mid-Output flowcell (300 cycles; 2 × 150 bp reads). The reads were trimmed with Trimmomatic (v0.36.5) [37]. The coverage obtained for all the sequenced phages are available in the Appendix A. The obtained data were assembled and annotated using PATRIC online platform (v3.6.2) [38] that employed SPAdes (v3.15.2) [39] and Pilon (v.1.23) for assembly, and RASTtk [40] and tRNAscan-SE [41] as annotation software. Default settings were applied.

The assembled genomes were compared with the NCBI databases using BLASTn. Manual functional annotation was performed by comparing PATRIC predicted ORFs against the non-redundant GenBank protein database [42] through BLASTp [43].

To exclude the presence of bacterial virulence genes, the phage genomes were screened against VirulenceFinder 2.0 database (v 2.0.3) (accessed on 20 August 2021) [44]. Pairwise global alignment between genomes was performed through EMBOSS Stretcher [45]. The online tool phageAI (https://phage.ai/) (accessed on 20 April 2020) [46] was used to predict phage lifestyles. Finally, the graphic visualization of the genomes and of their comparison was obtained with R package genoPlotR [47]. The variant calling analysis of highly similar phages was performed using the tool Snippy (v.4.6.0) with default settings through the Galaxy platform [48].

### 2.7. Phylogenetic Analysis

To define the evolutionary history of the new phage species, a phylogenetic analysis was performed. To choose the protein sequences to compare, we used as reference the protein used for the closest phage taxonomic group as indicated in the International Committee on Taxonomy of viruses (ICTV) documentation [49].

To generate a pphageB1 tree, the major capsid protein (MCP) and RNA polymerase (RNAPol) were used as reference genes to perform the analysis. For psageB1, only the MCP was used. Phage psageA1 was grouped and classified based on its DNA ligase and MCP.

The phages chosen for each tree were those included in the established ICTV trees used as reference, supplemented with other closely related phages found by BLASTp searches and tBLASTn comparisons to the NCBI protein database (accessed on 20 August 2021). The outliers were chosen between phages of close virus families according to ICTV. The alignments were performed using MAFFT through EMBL-EBI [50] and the trees were assembled with IQtree [51] using a maximum-likelihood method with a bootstrap number of 1000. The trees were visualized and graphically improved using the Mega7 software [52].

## 3. Results

### 3.1. Bacterial Isolation

Pph and Psa bacteria were successfully isolated from symptomatic leaves of bean plants and from symptomatic leaves and bark samples of kiwifruit plants in Piedmont, respectively. A total of 22 Psa strains, 10 Pph strains and nine strains of different *P. syringae* non-pathogenic to kiwifruit and bean (Table 1) were also considered. The PCR analysis performed with specific primers confirmed they were *P. syringae* belonging to the pathovar *actinidiae* or *phaseolicola*.

### 3.2. Phage Isolation and Morphology

Between 2018 and 2019, sixteen Pph and seven Psa phage isolates were obtained from soil collected from bean fields or kiwifruit orchards, respectively (Table 2). The phages were isolated by testing their ability to lyse Psa strain K7#8 or Pph strain Cuneo #6_18 (Table 1). No phages could be isolated from irrigation water or from plant symptomatic tissues. The phages isolated in the kiwifruit orchards displayed a variety in plaque size and morphology on the bacterial lawns. Conversely, all the phage isolates active against Pph strains caused similar plaques, clear and 5 mm in diameter, as described in Table 2. PsageK4e was isolated in the same area of psageK4, but from soil collected one year after the first sampling, and showed a similar plaque morphology.

TEM observations were performed on all the Psa phages and on some of the Pph phage isolates. These observations displayed capsids composed of icosahedral heads linked to tail structures, thus indicating their putative association to the *Caudovirales* virus order. All phages isolated in bean fields showed an icosahedral head and a short, stubby tail. PsageK4, psageK4e, and psageA1 had a typical myovirus morphology, with TEM images revealing sheathed contractile tails. Three isolated siphovirus-like phages (psageB1, psageK9 and psageB2) were characterized by their long flexible tail (Figure 1). PphageB1 phage had an isodiametric head with a diameter of 54 nm (SE = 0.4 nm; n = 84); psageA1 had an isodiametric head of 72 nm diameter (SE = 0.3 nm; n = 129) and a contractile tail of 125 nm length (SE = 0.8 nm; n = 101); psageB1 had a head of 78 nm diameter (SE = 0.4 nm; n = 83) and a flexible non-contractile tail of 176 nm length (SE = 2.2 nm; n = 74).

### 3.3. Phage Sequencing and Annotation

For groups of phages able to lyse the same subset of bacteria, and isolated from the same area, some representatives were chosen (15 total isolates) to extract and sequence their genomic DNA.

The phages subjected to whole-genome sequencing contained double-stranded DNA genomes with lengths ranging from 41.7 kb to 112 kb and a GC content ranging from 48.79% to 60.44%. In Table 3, the number of proteins encoded on the genomes of the sequenced phages are shown and were labelled as hypothetical proteins, ORFans (ORFs without sequence homologs in other known genomes), or proteins associated with a predicted function. Due to the methodology used for library synthesis associated with sequencing (Nextera Flex), the physical termini of our isolates were not able to be detected through bioinformatic methods. Thus, the sequences termini were arbitrarily based on the termini of the phages most similar in the NCBI database. Furthermore, to infer the type of genomic ends and packaging strategy, we followed the method of Merrill [53], and built a phylogenetic tree using the large terminases of psagaeA1, psageB1 and pphageB1, and the large terminases sequences of phages whose packaging systems and physical ends were experimentally determined (Appendix A). PsageA1 terminase was located with terminases of phages with 5′ *Cos* ends, psageB1 with phages with phages with 3′ *Cos* ends and pphageB1 with phages that have termini with short direct terminal repeats (DTRs). PsageA1, psageB1 and pphageB1 may share these packaging strategies. None of the phages showed evidence of toxins or bacterial virulence proteins encoded in their genomes. Proteins related to lysogeny were only found in psageB2. The online artificial intelligence platform PhageAI [45] agreed with our analysis, predicting a temperate lifestyle for psageB2 and a lytic lifestyle for all the other phages. Following the sequencing results the 15 isolates were divided into 5 groups (in Table 3 the characteristics of one representative for each group are displayed) based on the similarity of their sequences, and the molecular characteristic of each group is discussed below.

#### 3.3.1. Genomic Characteristics of pphageB1

The eight sequenced phages active against Pph (pphageB1, pphageB2_1, pphageT1_2, pphageT2_1, pphageBV2, pphageBV4, pphage BV7_1 and pphageBV7_2) showed an almost complete identical sequence at the nucleotide level and therefore phage pphageB1 was arbitrarily chosen as the type sequence.

PphageB1 sequence showed 73.89% similarity with *P. fluorescens* SBW25 phage phi-2 (NC_013638) but with a query cover of only 3%. Phi-2 was still used as the basis to assemble and orient the pphageB1 genome. The most similar phage to pphageB1 is MR18 with a sequence identity of 88.65% (query cover 51%). Nevertheless, the highly identical Pph phage isolates may be considered as representatives of a novel phage species [54]. The pphageB1 phage was assembled in one contig using 181,767 reads. The genome of pphageB1 was 41,714 bp long, with a GC content of 56.6% (Table 3). The pphageB1_1 genome codes for 52 open reading frames (ORFs), 25 of which encodes proteins with a predicted function. No integrases were found to be encoded on the pphageB1 genome, suggesting a lytic lifestyle.

A clear functional organization was noticed between two regions in the genome, the first associated with DNA metabolism (DNA replication, repair and recombination) and the second with morphogenesis genes. The small and large subunits of the terminase (pphageB1_44, pphageB1_45) involved in the packaging of the DNA and one protein associated with lysis (pphageB1_47) were located in the same region near the cluster of structural protein associated genes. No tRNA sequences were predicted in the pphageB1 genome. Notably, one of the structural proteins (pphageB1_39) had a C-terminal lysozyme fold having a putative double function: capsid formation and bacterial lysis.

In Figure 2A, the pphageB1 genome organization is compared with the sequence of phage MR18. The figure clearly displays the high level of synteny between the two phages, with a similar organization of clusters linked to the DNA processing and to the phage morphogenesis and lysis, although they have a lower grade of similarity in the regions associated with morphogenesis of the two genomes. PphageB1 has a holin encoding CDS (pphageB1_27) at the relative position where MR18 has a phosphodiesterase (PssvBMR18_gp31). The pphageB1 RNA polymerase(pphageB1_29) maintains its relative position, but with little similarity. The small and large subunits of the terminase (pphageB1_44, pphageB1_45), lysozyme (pphageB1_47) and Rz-protein (pphageB1_49) CDSs are in the same region. MR18 has an additional putative SGNA/GDSL protein (PssvBMR18_gp53) compared with pphageB1.

Regarding the other eight Pph phage isolates, four shared 100% similarity with a query coverage of 99% without detectable SNPs (pphageBV2, pphageBV4, pphageB1 and pphageBV71). Regarding the remaining four Pph phages, following a variant calling analysis through snippy software, the SNPs that differentiate pphageB1 from pphageB21, pphageT12, pphageT21, and pphageBV72 phages, were identified (Appendix A). PphageB2_1 had a majority of synonymous variants (119) and 28 missense variants. Only synonymous mutations were found in the morphology-associated protein named “unclassified head protein” (pphageB1_38) and in the phage terminase subunits (pphageB1_44, pphageB1_45). The missense SNPs were located in a limited number of ORFs associated with (I) structural functions (predicted phage tail tubular protein A pphageB1_36, predicted phage tail tubular protein B pphageB1_37), (II) DNA metabolism (two phage exonucleases pphageB1_23 and pphageB1_25), (III) lysis (a predicted phage lysozyme pphageB1_47 and a peptidoglycan lytic exotransglycosilase pphageB1_40), and (IV) in five ORFs coding for hypothetical proteins. PphageT21 compared with pphageB1 had 110 synonymous variants mostly in structural proteins and in the DNA polymerase coding sequence; 28 missense variants were found in protein associated with morphology, DNA metabolism and lysis and in six hypothetical proteins. PphageT12 had 680 synonymous changes and 67 missense mutations, with the majority of the missense SNPs concentrating in the lysis proteins and in the phage tail fibers (pphageB1_41, pphageB1_42). PphageBV72 had 69 synonymous variations and 14 missense changes with a distribution in proteins similar to the other Pph phages SNPs compared with pphageB1 genome; additionally, there was a missense mutation in the DNA ligase protein sequence (pphageB1_16).

#### 3.3.2. Genomic Characteristics of psageB1

PsageB1 showed 83.6% identity along 72% of the genome of *P. syringae* pv. *avii* phage nickie (NC_042091.1) (Figure 2B), isolated from wastewater. A global alignment performed with EMBOSS Stretcher revealed an identity of 72%. This was below the identity threshold to distinguish phages at the species level (95%). Therefore, psageB1 may be considered a representative of a novel species of phages infecting Psa. The phage had a genome of 112,269 bp with a 56.47% GC content and contained 161 ORFs of which the majority of encoded proteins were classified as hypothetical proteins (105) and ORFans (16). Only one fourth of the ORFs were associated to proteins with a predicted function (40) (Table 3). Four tRNA sequences were found in psageB1, organized in a small cluster.

As shown in Figure 2, the genome organization is similar to phage nickie, with a region associated to morphogenesis genes, followed by a cluster of genes linked to lytic activity (holin psageB1_039, lysozyme R psageB1_040, two GDSL-like lipases psageB1_034, psageB1_038 and a particle-associated lyase psageB1_035) and a series of genes linked to DNA processing (Figure 2B).

In the arbitrarily assigned terminal part of the sequence, both phages were annotated with a long series of short hypothetical proteins that shared little identity between the two phages. Both phages had a similar large terminase (psageB1_012_) in the same genomic region. Nickie had an additional GDSL-like lipase (CNR34_00036) next to the holin ORF shared with psageB1 (psageB1_039). PsageB1 had an ORF associated with the recombinase A (recA) (psageB1_066) and two ORFs associated with recombination related exonucleases (psageB1_069, psageB1_070). It seems that recA-related genes are used in replication in *Siphoviridae* [55] and *Myoviridae* [56] members.

#### 3.3.3. Genomic Characteristics of psageA1

Phages psageA1 and psageA2 shared 100% identity at the nucleotide level. The psageA1 and psageA2 sequence showed partial similarity (74.28% identity with 10% query cover) with the lytic *P. syringae* pv. *avii* phage ventosus (HMG018930). These phages may be considered as representatives of a new species active against Psa following ICTV guidelines. The total length of the psageA1 genome was 98,780 bp and it had a 48.79% GC content. Annotation (Figure 2C) found 176 different ORFs, 51 of which encoded a predicted function. The remaining ORFs were associated with hypothetical proteins (76) and ORFans (16) (Table 3). The genome organization was similar, although with very limited sequence similarity to the organization of the *Otagovirus* genus [16] with a tRNA cluster followed by the large terminase protein (psageA1_065), a region with the capsid, tail and baseplate structural components (psageA1_066, psageA1_067, psageA1_069, psageA1_070, psageA1_072, psageA1_074, psageA1_075, psageA1_078, psageA1_080, psageA1_083, psageA1_084), followed by a lysin (psageA1_092) and a region with all the proteins used for DNA replication. In psageA1 there were 14 different sequences coding for tRNAs. Additionally, psageA1 seemed to have a tRNA-specific adenosine deaminase (psageA1_042) (Figure 2C).

#### 3.3.4. Genomic Characteristics of psageB2 and psageK9

PsageB2 and psageK9 pairwise alignment showed high identity (99.5%). This sequence showed an identity of 99.95% along 97% of its sequence with the *P. syringae* pv. *actinidiae* phage phiPsa1 [15] (KJ507100) (Figure 2E). This identity was greater than 95%, thus psageB2 cannot be considered as a new phage species, but it was an isolate of the phiPsa1 species [54]. The psageB2 dsDNA genome was 50 kb long, with a GC content of 58.51%. Only 26 of the 77 identified ORFs had a predicted function assigned for their product, and the majority of the rest of ORFs were annotated as hypothetical proteins (47) and ORFans (4). No tRNA sequence was found in psageB2 genome by tRNAscan-SE (Table 3). The genome organization was the same as phiPsa1 with a clear distinction of regions assigned to morphogenesis followed by an endolysin ORF and DNA replication- and modification-related ORFs followed by one single ORF associated to the large terminase (psageB2_). The sequence of both psageK9 and psageB2 had a CDS of an integrase (psageB2_018c) and of a cI repressor (psageB2_043c) that identified them as lysogenic phages, as was confirmed for phiPsa1. Furthermore, two recombinase genes were annotated, NinB and NinG (psageB2_054 and psageB2_056).

#### 3.3.5. Genomic Characteristics of psageK4

Phage isolates psageK4 and psageK4e had a 99% identity between each other. The psageK4 sequence had a 96.84% identity with 97% coverage of the sequence of the Psa phage phiPsa267 [16] (MT670417.1) (Figure 2D). The psageK4 assembly gave a sequence of 98,440 bp with a GC percentage of 60.44%. Among its numerous annotated ORFs (179), the majority were classified as hypothetical proteins (112) and ORFans (16), and only 51 ORFs had a specific function assigned following the annotation pipeline (Table 3). Eighteen tRNAs were organized for the main part in a cluster positioned similarly to psageA1. A psageK4 and psageK4e genomes variant calling analysis (Appendix A) revealed the presence of 836 synonymous SNPs prevalently on structural proteins and 159 missense SNPs on structural protein also but on the terminase (psageK4_064), DNA polymerase(psageK4_115), and phosphoesterase (psageK4_129), in addition to many hypothetical proteins associated with ORFs. A BLASTn comparison of the two genomes (Figure 2D) revealed the presence of an additional ORFan in psageK4e between the ORFs similar to psageK4_102 and psageK4_103, and the absence of two ORFans present in psageK4 (psageK4_164 and psage_138).

PsageK4 and psageK4e had the same genome organization of phages of the genus *Otagovirus*. This structure, despite the limited similarity shared, was similar to that previously described for psageA1 as displayed comparing Figure 2C,D.

### 3.4. Phage Host Ranges

Ten of the fifteen sequenced phages were selected to be tested for their ability and efficiency to lyse with an EOP assay the strains of Pph and Psa present in the collection. The selection of phages for this assay was based on having at least one phage representative for each of the five groups identified through sequencing and to account for the diversity within these groups. As shown in Table 1, pphageB1 was able to lyse every strain of its target bacteria, but was ineffective in producing plaques on any Psa strain. The isolates with genomes highly similar to pphageB1, pphageB21, pphageBV72, pphageT12 and pphageT21 also showed a similar host range. Conversely, some of the phages isolated from kiwi orchards showed the ability to lyse several Pph isolates in addition to their ability to lyse many if not all of the collected Psa strains. The psageK4e isolate was the most effective against both *P. syringae* pathovars in the collection. Notably, psageK4 isolate could lyse all the Psa and Pph strains in the collection but displayed low efficiency in the case of Pph strains. Psagek4e had a similar host range but was also able to effectively infect Pph strains. PsageA1 and psage B2 were not efficient or not able at all to lyse Pph strains. To expand the possibility of a cocktail to use in phage therapy application, we tested the specificity of another phage collection we had access to [13] on our Psa and Pph bacterial strains: only one of the phages, MR8, a phage similar to pphageB1, was able to efficiently infect the strains in the collection (Table 1). MR8 was able to infect both the pathovars of *P. syringae*, but had a greater EOP on the Pph strains. Phages MR1 and MR2 had 90% identity with the phage PPPL-1 (NC_028661), capable to lyse Psa, but could not lyse any of our collection Psa strains (not shown).

All phages showed a reduced activity against the *P. syringae* non-pathogenic strains tested. Despite this, psageB1 was able to lyse three nonpathogenic strains efficiently.

### 3.5. Phage Resistance to Abiotic Stresses

A representative for further assays was selected for each of the five groups (clade-like) described above, excluding the group with predicted lysogenic properties which are not of immediate exploitation for phage therapy for plant diseases. Nevertheless, temperate phages could be used as biocontrol agents with different approaches described previously [57], and we do not exclude finding an application in biocontrol for psageB2 and psageK9 in the future.

Figure 3 displays the ability of pphageB1, psageK4, psageA1 and psageB1 to conserve infectivity after being exposed to three abiotic stresses: pH, temperature and UV-C irradiation. After 18 h of exposition to different pH environments, phages showed a similar behavior, keeping their activity stable between pH 4 and 10 with the exception of psageB1 that reduced its titer to 99% at pH 10 compared with pH 4. All of phages rapidly decreased in their capacity to lyse at the most acidic pH tested, pH 2 (Figure 3A). Regarding exposure to heat stress, all tested phages showed a comparable titer when exposed to 4 °C, 26 °C and 37 °C for 1 h (Figure 3B). However, at 55 °C, no viable phage particles could be detected for the phages psageB1 and pphageB1. At the same temperature, the psageK4 virus titer remained stable after one hour, and psageA1 showed a reduced activity, but not a complete inactivation (Figure 3B). The irradiation with UV-C caused a constant decrease in the number of infectious phages counted after exposure, already visible after 10 min. In the majority of cases, a complete inactivation of the phage particles was observed after two hours. Notably, one of the phages tested, pphageB1, showed the ability to lyse its target bacteria even after 3 h of exposure (Figure 3C).

### 3.6. Phylogenetic Analysis

The three phage representatives of new species (pphageB1, psageA1 and psageB1) were classified in the *Caudovirales* order following our phylogenetic analysis. The phylogenetic trees built comparing the major capsid protein (MCP) (Figure 4A) and RNA polymerase (Appendix A) of pphageB1 revealed that this virus is likely belonging to the *Autographiviridae* family and *Krylovirinae* subfamily. *Autographiviridae* in the past was a group included in the *Podoviridae* family, because of similar capsid morphology. PphageB1 seems to cluster together with the clade 3 of phages outlined by Rabiey et al. (2020) from which we used MR5 as a representative. PphageB1 and MR5 seem to be not similar enough to share a genus but are likely to be located in the same subfamily. Two phylogenetic trees based on MCP (Appendix A) and DNA ligase of psageA1 (Figure 4B) identify the phage as an unclassified myovirus positioned near the genus of phiPsa267, the genera *Otagovirus* and *Pakpunavirus*. The phylogenetic tree comparing the MCP of psageB1 (Figure 4C) with the closest phages present in databases, locates the phage in the *Nickievirus* genus and suggests the phage as being a member of a new species of this genus.

## 4. Discussion

Bleeding canker of kiwifruit and halo blight of bean cause worldwide significant crop losses [2,58]. Strategies to prevent and cure these pathologies without the use of antibiotics or copper-based pesticides are currently not available. An IPM approach that includes the use of bacteria’s natural predators and parasites—specifically bacteriophages—as biological control, could be an alternative method that can be adopted to minimize chemical control. In particular, for biocontrol purposes, lytic phages are required to have a high stability in the environment. Phages have the advantages of being very specific and not leaving toxic residues on the plant.

One of the most numerous Psa phage collections was gathered in New Zealand in 2014 [16] with 258 phages active against Psa, of which 24 were observed by TEM, revealing myovirus-, podovirus- and siphovirus-like morphologies. Among these phages, myovirus phiPsa21 showed an unusually large genome and was later recognized as a jumbo phage [59], containing genes possibly associated with the formation of a nucleoid-like structure in the host cell. In the same year, Di Lallo [15] isolated and sequenced two Psa phages: phiPSA1 and phiPSA2. While phiPSA2 displayed a good efficacy against the Psa strain tested, phiPSA1 showed a temperate lifestyle and a narrower host range. The group of Flores [21] isolated and sequenced four *Podoviridae* phages similar to phiPsa2 [15]: CHF1, CHF7, ChF19 and CHF21; these phages were tested alone and combined as biocontrol agents and showed a significant decrease in Psa symptoms in vitro and a decrease in the bacterial load on kiwifruit plants in a greenhouse trial. In Korea, Yu and co-authors characterized two myoviruses and three podoviruses active against Psa, which were tested for their stability to different pH and UV-B exposition but were not tested as biocontrol agents [18]. Yin and collaborators in 2019 isolated 36 phages from surface water samples in the Shanghai region, using the strain of Psa biovar 3 XWY0007 as target bacteria [22]. The phages were represented mainly by myovirus-like particles, with some phages similar to *Podoviridae* and *Siphoviridae* families. In summary, this shows that there are a large variety of Psa phages with a certain degree of overlapping features from different geographical locations.

A much lower variety and number of Pph phages are reported in literature, with the already cited cystovirus phi6 being the first isolated Pph phage [23,25]. Later, two more phages were isolated, but they were not characterized molecularly, one with a podovirus-like morphology and one with a cystovirus morphology [24]. Phi6 was tested as a biocontrol agent against *P.s*. pv. *syringae* [14] and against Psa [19], with a decrease in the titer of two strains of Psa biovar 3 in vitro and with a minor effect of reduction in the bacteria number in ex planta experiments performed on excised kiwifruit leaves. However, the efficacy of phi6 against Pph strains has not yet been assessed.

The current study isolated phages from soil in the same areas of plants infected by Psa or Pph, but was not able to isolate viruses from water sources or from symptomatic tissue, indicating a possible greater stability of phages in soil as indicated by Iriarte [60].

All the virus isolates this current study found belong to the order *Caudovirales*, consistent with more than 97% of all described *Pseudomonas* phages [61]. No genes associated to lysogeny were found, except for psageB2.

The psageK4 isolate resulted in belonging to the same species of virus phiPSA267 [19] isolated in New Zealand, while psageB2 belonged to the same species as phiPSA1 [15], and M13 and M15 [13], isolated in Italy and England, respectively. PsageB2 is temperate just like phiPSA1 and has a narrow host range compared with the other phages isolated by our group. The isolation of phage variants belonging to the same species is probably due to the global diffusion of Psa biovar 3 strains [62] and consequently the global presence of the viruses associated with it.

The virus isolate psageK4e was collected in the same location as psageK4 one year later, and shares with it a similarity of 99%, indicating a consistence of phage population in the same field in distinct growing seasons. The two phage genomes are highly similar in the region where all the genes linked to the morphology are located, the DNA terminase and some of the ORFs linked to DNA metabolism. They are less similar in their regions with a great number of ORFs encoding hypothetical proteins. A limited number of ORFs coding for hypothetical proteins are present only in one of the two phages. Their high similarity is reflected in their capability to lyse almost the totality of Psa and Pph strains of our collection. However, the EOP of psageK4e is significantly greater than psageK4 EOP.

Phage psageA1 can be considered a representative of a novel species and it may represent the only member of a new genus of *Myoviridae* with the most similar phages located in the *Otagovirus* genus. For this new genus, we propose the name ‘Mantavirus’ and accordingly the name ‘Mantavirus psageA1′ for the species including our isolates psageA1 and psageA2.

Phage psageB1 can be described as a representative of a new species of *Siphoviridae* that can be situated within the *Nickievirus* genus. Thus, we propose the binomial name ‘Nickievirus psageB1′ for this species. Its genome codes for a recombinase, RecA, but this is probably not linked to a temperate lifestyle or transduction, but to DNA replication as in other phages [55]; in fact, a temperate behavior was not observed in our host range. This protein can also have a role, such as in other siphoviruses, in the evolution through recombination of the phage [63,64].

Based on these results, PphageB1 in combination with the *P. syringae* non-pathogenic strains SZ131, UB210 and UB246 can be used for future experiments in a carrier bacteria approach [65] to sustain the population of one of the phage or both on the phyllosphere of kiwifruit and beans in a possible biocontrol approach.

Pph is an endemic pathogen in Piedmont and was isolated in Italy in 1962 [66]. All Pph phage isolates of Piedmont and Veneto were associated through sequencing to the same novel virus species. The isolation in different places of almost identical phages in the same species may be linked to a homogeneous population of Pph in northern Italy, which could be race 6, the most prevalent worldwide [4]. The Pph phages variant call analysis revealed that the majority of changes were synonymous mutations, the main part of which were located on structural genes whose sequence is bound to the stability of the capsid, which are highly sensitive to even small changes [67]. Consistently, the main part of the morphology-associated genes is not subject to variation as many genes are linked to DNA metabolism. The missense mutations are located mainly on the phage tail fiber, and on the proteins of the lysis cassette suggesting differences in their host range in comparison with pphageB1. These mutation hotspots may be crucial for the Pph phages to continuously adapt to their bacterial host in a process of coevolution. The variation occurred mainly between the phages isolated in Piedmont, with a minor diversity between the Piedmont phages and the Veneto phages and inside the group of Veneto phages. SNPs in very similar phages were reported to be linked in differences in stability to abiotic stresses and lysis time [68], so these phages, although very similar, can be used to design phage cocktails.

Concerning the pphageB1 sequence, both its high genome sequence similarity to existing phages and its genome organization, support its assignment to the *Autographiviridae* family in the *Krylovirinae* subfamily. Its genome sequence is not similar enough to any phage in the genera now described within *Krylovirinae*, so we propose this isolate with the pphageB21, pphageBV72, pphageT12 and pphageT12, as representatives of a new genus called ‘Ceppovirus’, containing only one species, with the proposed binomial name “Ceppovirus pphageB1′. Like the other members of this family, pphageB1 encodes for a single-subunit DNA-directed RNA polymerase (pphageB1_29), placed in proximity of the cluster of DNA processing genes [69]. Interestingly, one of the structural proteins (pphageB1_39) has a lysozyme C-terminal domain that can be linked to bacterial lysis, even if the target bacteria is Gram-negative [70]. This enzyme is likely linked to the lysis of biofilm polymer or to expose the phage receptor on the bacteria surface degrading the peptidoglycan. PphageB1 has a similar genome organization in comparison with phage phiPSA17 [17] and the *P. syringae* phages Bertil, Misse and Strit [71]. PphageB1 sequence has no strong homologies to other phages in literature, and shows the highest identity with *P. morsprunorum* phage MR18. Furthermore, pphageB1 shares a limited similarity with the *P. fluorescens* phage phi-2: pphageB1 phage should be tested in the future for the capability to lyse *P. fluorescens* strains indicated by Frampton [16,17] since it could be a possible carrier bacteria for biocontrol. A carrier strain would protect the phages from UV radiation and increase the phage number during the application.

A great number of tRNA sequences in phages is often associated with a lytic lifestyle and a greater host range because of the higher adaptive potential to different host codon usages and to different composition of the bases in genomes that can be partially indicated by the GC content [72,73]. Our isolate psageA1 has a GC content (48%) much lower than Psa (58%) [74] and the presence of 14 tRNA sequences in psageA1 could partially explain its wide host range in our Psa and Pph collection. Conversely, psageB2 (GC content of 58%), psageB1 (GC content of 56%) and pphageB1 (GC content of 56%) have a more similar GC content to the Pph GC percentage (60%) [75] and this is consistent with the presence of only four tRNA in psageB1 and of no tRNA in psageB2 and pphageB1.

Bacteriophage biocontrol in agriculture to control phytobacterial diseases faces several challenges such as scarce persistence in the plant phyllosphere, high diversity of the target and high probability of resistance development [9]. Possible solutions are the use of protective formulations, development of phage propagating bacterial strains and adapting timing and frequency of application [76].

The isolated phages performed well in vitro under the tested stress conditions. These experiments may be indicative of their stability to stress occurring in the open field, but confirmatory experiments in field conditions are required. Phages psageK4, psageA1, psageB1 and pphageB1 resist for one hour temperature ranges normally found in the field on the leaf surface, so they should be applied at dawn to avoid the effects of the temperature and UV radiation, and increase their persistence on the phyllosphere. Most phages are resistant to a pH from 5 to 8 [77]; our phages were only inactivated at pH 2, but their infectivity was not affected by an 18 h exposure to pH from 4 to 10. This may have an important role for their possible application on kiwifruit vines through trunk injection as the pH in the xylem lies around 5. Additionally, the phages we collected would not be affected by the pH of the majority of formulants, that are not highly acidic [77].

Phage cocktail persistence in the open field could be improved with specific formulations [9] and their action can be increased in synergy with the addition with other compounds such as carvacrol [20] and garlic extract [24]. Additionally, the phages we isolated can be a useful source of endolysins as an alternative Psa and Pph biocontrol method [78]; the vast number of functionally uncharacterized ORFs they encode could contribute to further control approaches once their role is elucidated, urging a vast effort to assign function to the unprecedented proteomic diversity harbored by phages [67,79,80].

As reported previously in literature [21,23] UV exposure is detrimental to most phages. In our assays, the majority of the phage collection indeed showed sensitivity to UV light. However, the phage pphageB1 has a greater resistance to this treatment, increasing its possible persistence on the phyllosphere of plants in a biocontrol setting. This is important in relation of the high amount of UV radiation the plants’ surface is exposed to in kiwifruit orchards.

Host range analysis showed a versatility of all the Psa phage isolates that were able to lyse many of the Pph strains of our collection. Conversely, Pph phages were not capable to lyse any of the Psa strains. Thus, the Psa phages that we isolated may recognize receptors shared by the P.s. species complex phylogroup of Psa (PG1) and Pph (PG3) [81]. To better elucidate the nature of these receptors, adsorption analysis on both bacterial phytopathogens will be necessary. It is possible that the bacteria belonging to pathovar *actinidiae* and *phaseolicola* also share the replication and translation bacterial machinery involved in a successful phage infection. The *P. syringae* pv. *morsprunorum* and *syringae* phage MR8 [13] was able to lyse several Psa strains and one Pph strain, making it a suitable candidate to be inserted in a future cocktail designed against both phytobacteria subjects of this study. *P*. *syringae* pv. *morsprunorum* race 1 is located in the same phylogroup of the P.s species complex of Psa (PG1), and *P. syringae* pv. *morsprunorum* race 2 in the same phylogroup as Pph (PG3), supporting the hypothesis of shared susceptibility determinants between these two phylogroups.

The isolated phages will be tested in different combinations or cocktails allowing a broader host range and a possible synergy between phage actions [82]. Psa phages have been reported to infect other *P. syringae* pathovars such as *morsprunorum* [83] and other *Pseudomonas* species such as *P. fluorescens* [77]. The phages we isolated can be active against both Psa and Pph similarly to the phages isolated by Yin [22], broadening the possibility to design a cocktail against these phytobacteria. Therefore, they could potentially be useful to design cocktails against other phytobacteria in the *P. syringae* species complex.

## Figures and Tables

**Figure 1 viruses-13-02083-f001:**
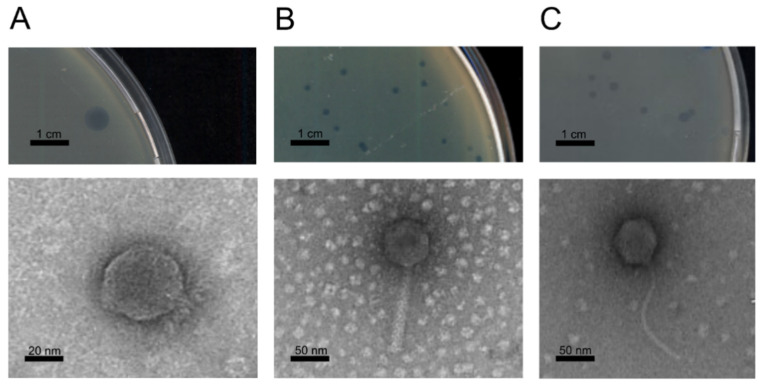
Plaque morphology and electron microscopy images of the three phages representing putative new species: (**A**) pphageB1 capsid has an icosahedric head with a short tail and clear fibers, typical for a podovirus morphology; (**B**) psageA1 has an icosahedral head and a putative contractile tail typical for the myovirus morphology; (**C**) psageB1 shows a long and flexible non-contractile tail linked to an icosahedral head, a typical siphovirus morphology.

**Figure 2 viruses-13-02083-f002:**
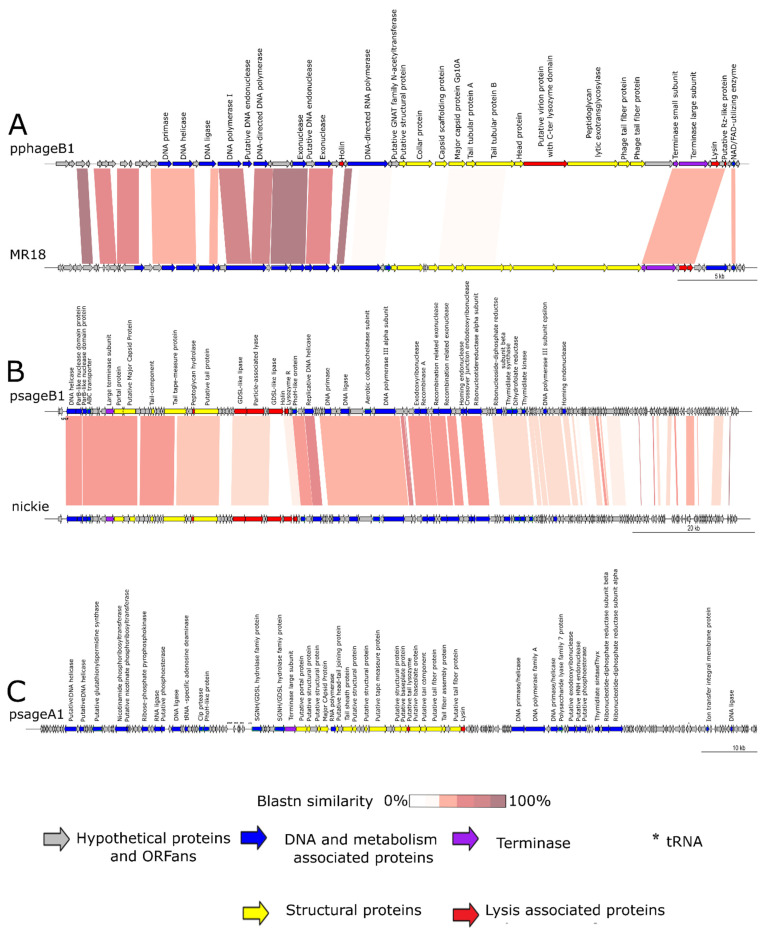
Genome organization of pphageB1 phage compared with MR18 (**A**), psageB1 compared with nickie (**B**), psageA1 (**C**), psageK4 compared with psageK4e and phiPsa267 (**D**), and psageB2 compared with phiPsa1 (**E**). The arrows indicate annotated ORFs; and the asterisks, annotated tRNA sequences. Yellow arrows indicate ORFs associated with structural proteins, blue DNA- and metabolism-associated ORF, red ORFs that encode for lysis associated proteins, and purple ORFs homologues to terminases. The intensity of the color between two compared sequences indicates percentages of BLASTn similarity.

**Figure 3 viruses-13-02083-f003:**
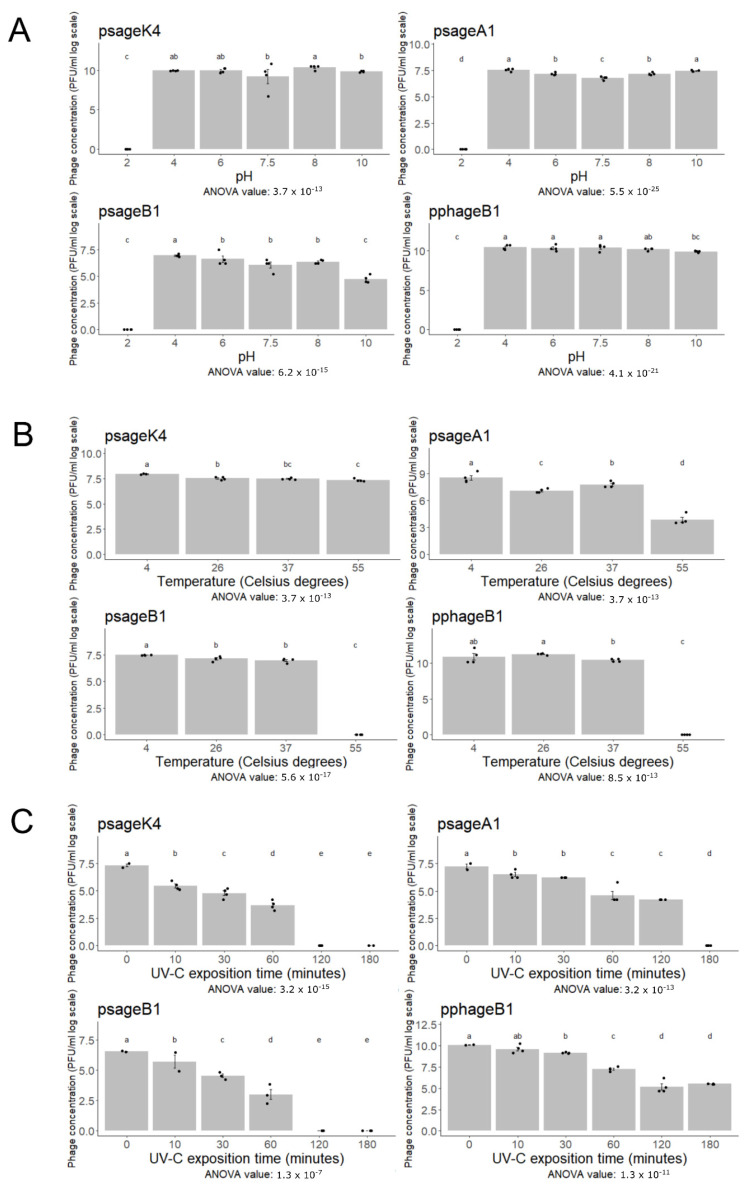
The charts show the viability of pphageB1, psageK4, psageB1 and psageA1 phages under exposure to different pHs (**A**), or temperatures (**B**), for one hour, and to different times of UV-C irradiation (**C**). The number of replicates for every condition is four, and the error bars indicate the standard error. The statistical significance was calculated through the agricolae R package and consists of ANOVA (ANOVA *p*-values are shown under every plot) and Kruskal–Wallis analysis (significant differences indicated by letters over the bars). Different lower-case letters above each bar plot represent statistically different values.

**Figure 4 viruses-13-02083-f004:**
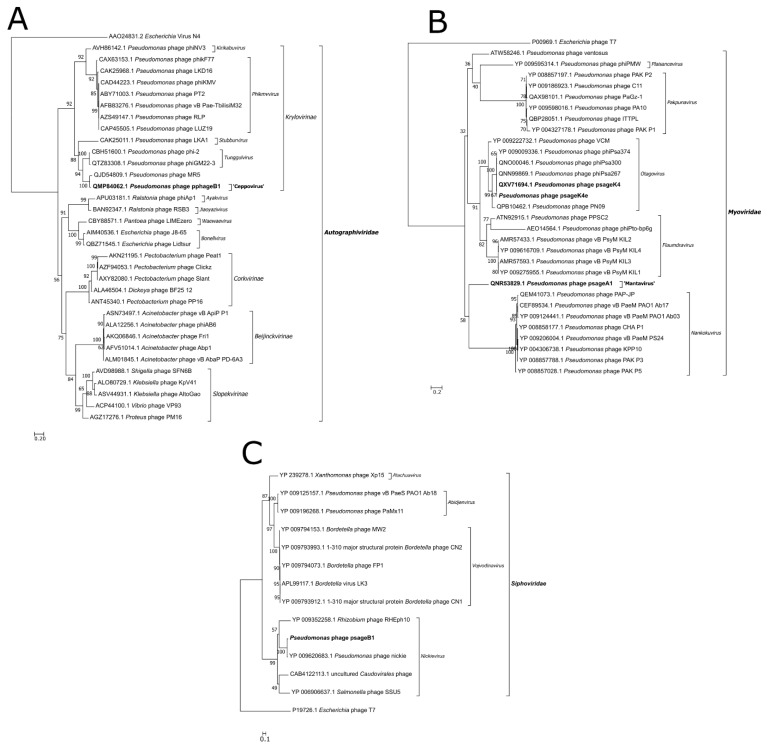
Phylogenetic analysis of a selected number of Major Capsid Protein (MCP) sequences similar to the MCP of pphageB1 phage (**A**), of a selected number of DNA ligases similar to the DNA ligase of psageA1, psageK4 and psageK4e phages (**B**), and of a selected number of MCP similar to the psageB1 MCP (**C**). The *Escherichia* virus T7 MCP and DNA ligase were used as an outgroup for psageB1 and psageA1. The N4 phage MCP sequence was used as an outgroup for pphageB1 tree. The maximum likelihood methodology was used to obtain the best tree. After comparisons, the chosen best model of substitution was WAG+G4 for the pphageB1 and psageB1 trees and Blosum62+G4 for the psageA1 tree. Consensus trees were constructed from 1000 bootstrap trees. PphageB1, psageA1 and psageB1 are indicated in bold characters. At the nodes, the bootstrap values, expressed in percentages, are indicated.

**Table 1 viruses-13-02083-t001:** *Pseudomonas syringae* pv. *actinidiae* and pv. *phaseolicola* strains properties and their phage sensitivity profiles. The key to phage efficiency of plating (EOP) is shown at the right end of the table. Red color indicates an EOP comparable to the one measured in the isolation strain (K7#8 for psageA1, psageK4, psageK4e, psageB1 and psage B2; Pph Cuneo #6_18 for pphageB1, pphageB21, pphageBV72, pphageT12 and pphageT21; Pph Cuneo #6_18 is arbitrarily chosen as reference for MR8). The other colors represent different levels of EOP with darker colors indicating a greater efficiency in their lytic activity. “-“ symbols represent cases for which plaques have not been observed at any concentration of the crude lysate. * Variety *deliciosa*; ^$^ Phytosanitary Inspection Service.

Identification	Origin	Geographical Origin	Source	Bacterial Strains	Year	pphageB1	pphageB21	pphageBV72	pphage T12	pphage T21	psage A1	psage K4	psage K4e	psage B1	psage B2	MR8		
*P.s.* pv. *actinidiae*	*Actinidia chinensis* *	Piedmont	Agrion	K7#1	2019	-	-	-	-	-								EOP
*P.s.* pv. *actinidiae*	*Actinidia chinensis* *	Piedmont	Agrion	K7#8	2019	-	-	-	-	-								**<0.01**
*P.s.* pv. *actinidiae*	*Actinidia chinensis* *	Piedmont	Agrion	K4#3	2018	-	-	-	-	-								**0.01–0.1**
*P.s.* pv. *actinidiae*	*Actinidia chinensis* *	Piedmont	Agrion	K4#6	2018	-	-	-	-	-	-				-			**0.1–1**
*P.s.* pv. *actinidiae*	*Actinidia chinensis* *	Piedmont	Agrion	K4#7	2018	-	-	-	-	-								**1**
*P.s.* pv. *actinidiae*	*Actinidia chinensis* *	Piedmont	Agrion	K4#9	2018	-	-	-	-	-	-							**1–10**
*P.s.* pv. *actinidiae*	*Actinidia chinensis* *	Piedmont	Agrion	K4#10	2018	-	-	-	-	-	-				-			**10–100**
*P.s.* pv. *actinidiae*	*Actinidia chinensis* *	Veneto	PIS ^$^	#5712.19	2019	-	-	-	-	-								** >100 **
*P.s.* pv. *actinidiae*	*Actinidia chinensis* *	Veneto	PIS ^$^	#5726.19	2019	-	-	-	-	-	-				-			
*P.s.* pv. *actinidiae*	*Actinidia chinensis* *	Veneto	PIS ^$^	#5747.19	2019	-	-	-	-	-								
*P.s.* pv. *actinidiae*	*Actinidia chinensis* *	Veneto	PIS ^$^	#5846.19	2019	-	-	-	-	-	-				-			
*P.s.* pv. *actinidiae*	*Actinidia chinensis* *	Veneto	PIS ^$^	#5847.19	2019	-	-	-	-	-								
*P.s.* pv. *actinidiae*	*Actinidia chinensis* *	Veneto	PIS ^$^	#5850.19	2019	-	-	-	-	-								
*P.s.* pv. *actinidiae*	*Actinidia chinensis* *	Piedmont	PIS ^$^	#298a	2019	-	-	-	-	-	-				-			
*P.s.* pv. *actinidiae*	*Actinidia chinensis* *	Piedmont	PIS ^$^	#391a	2018	-	-	-	-	-								
*P.s.* pv. *actinidiae*	*Actinidia chinensis* *	Piedmont	PIS ^$^	#392a	2018	-	-	-	-	-				-				
*P.s.* pv. *actinidiae*	*Actinidia chinensis* *	Piedmont	PIS ^$^	#453d	2016	-	-	-	-	-	-				-			
*P.s.* pv. *actinidiae*	*Actinidia chinensis* *	Piedmont	PIS ^$^	#453e	2016	-	-	-	-	-				-				
*P.s.* pv. *actinidiae*	*Actinidia chinensis* *	Piedmont	PIS ^$^	#454d	2018	-	-	-	-	-	-				-			
*P.s.* pv. *actinidiae*	*Actinidia chinensis* *	Piedmont	PIS ^$^	#454e	2018	-	-	-	-	-	-				-			
*P.s.* pv. *actinidiae*	*Actinidia chinensis* *	Piedmont	PIS ^$^	#509c	2018	-	-	-	-	-								
*P.s.* pv. *actinidiae*	*Actinidia chinensis* *	Piedmont	PIS ^$^	#509d	2018	-	-	-	-	-								
*P.s.* pv. *actinidiae*	*Actinidia chinensis* *	Piedmont	PIS ^$^	#509e	2018	-	-	-	-	-								
*P.s.* pv. *phaseolicola*	*Phaseolus vulgaris*	Piedmont	PIS ^$^	Pph PSS	2015						-		-		-	-		
*P.s.* pv. *phaseolicola*	*Phaseolus vulgaris*	China	Private company	Pph #9	2015										-			
*P.s.* pv. *phaseolicola*	*Phaseolus vulgaris*	China	Private company	Pph #13	2015						-				-			
*P.s.* pv. *phaseolicola*	*Phaseolus vulgaris*	China	Private company	Pph #14	2015						-				-			
*P.s.* pv. *phaseolicola*	*Phaseolus vulgaris*	Piedmont	IPSP	Pph Cuneo	2016						-				-	-		
*P.s.* pv. *phaseolicola*	*Phaseolus vulgaris*	Piedmont	Agrion	Pph Cuneo #6_17	2017						-				-	-		
*P.s.* pv. *phaseolicola*	*Phaseolus vulgaris*	Piedmont	Agrion	Pph Cuneo #6_18	2018						-				-			
*P.s.* pv. *phaseolicola*	*Phaseolus vulgaris*	Piedmont	Agrion	Pph Cuneo #7	2018						-				-			
*P.s.* pv. *phaseolicola*	*Phaseolus vulgaris*	Piedmont	Agrion	Pph Cuneo #13	2017						-				-	-		
*P. syringae*	Water	France	INRAe	UB197	2010	-	-	-	-	-	-	-	-	-	-	-		
*P. syringae*	Water	France	INRAe	UB210	2010	-	-	-	-	-	-	-	-		-	-		
*P. syringae*	Water	France	INRAe	UB246	2010	-	-	-	-	-	-	-	-		-	-		
*P. syringae*	Water	United States	INRAe	USA052	2010	-	-	-	-	-	-	-	-		-	-		
*P. syringae*	Water	France	INRAe	CC1504	2010	-	-	-	-	-	-	-	-	-	-	-		
*P. syringae*	Water	France	INRAe	SZ030	2010	-	-	-	-	-	-	-	-	-	-	-		
*P. syringae*	Water	France	INRAe	SZ122	2010	-	-	-	-	-	-	-	-	-	-	-		
*P. syringae*	Water	France	INRAe	SZ131	2010	-	-	-	-	-	-	-	-		-	-		
*P.s.* pv. *syringae*	*Prunus armeniaca*	Lombardia	UNIMI	Pss ML#1	2018	-	-	-	-	-	-	-	-	-	-	-		

**Table 2 viruses-13-02083-t002:** Summary of the main features of phages isolated in this study. The classification of the phages is based on their phylogeny (Figure 4).

Phage Name	Source	Origin ^a^	Field Species and Cultivar	Source	Region	Year	Plaque Morphology (LB_LS_ 0.6%) ^b^	Classification ^c^
psageK4	Soil	Manta (CN)	*Actinidia chinensis* var*. deliciosa*	Agrion	Piedmont	2018	Clear, 1 mm	*Myoviridae*
psageK4e	Soil	Manta (CN)	*Actinidia chinensis* var*. deliciosa*	Agrion	Piedmont	2019	Clear, 1 mm	*Myoviridae*
psageK9	Soil	Borgo d’Ale (VC)	*Actinidia chinensis* var*. deliciosa*	IPSP	Piedmont	2019	clear or turbid 2 mm	*Siphoviridae*
psageA1	Soil	Manta (CN)	*Actinidia chinensis* var*. deliciosa*	Agrion	Piedmont	2018	Clear, 1–2 mm	*Myoviridae*
psageA2	Soil	Manta (CN)	*Actinidia chinensis* var*. deliciosa*	Agrion	Piedmont	2018	Clear, 1–2 mm	*Myoviridae*
psageB1	Soil	Manta (CN)	*Actinidia chinensis* var*. deliciosa*	Agrion	Piedmont	2018	Clear, 2 mm	*Siphoviridae*
psageB2	Soil	Manta (CN)	*Actinidia chinensis* var*. deliciosa*	Agrion	Piedmont	2018	Clear or turbid, 2 mm	*Siphoviridae*
pphageB1	Soil	Boves (CN)	*Phaseolus vulgaris* cv. *Billò*	Agrion	Piedmont	2018	Clear, 5 mm	*Autographiviridae*
pphageB12	Soil	Boves (CN)	*Phaseolus vulgaris* cv. *Billò*	Agrion	Piedmont	2019	Clear, 5 mm	*Autographiviridae*
pphageB13	Soil	Boves (CN)	*Phaseolus vulgaris* cv. *Billò*	Agrion	Piedmont	2020	Clear, 5 mm	*Autographiviridae*
pphageB21	Soil	Boves (CN)	*Phaseolus vulgaris* cv. *Billò*	Agrion	Piedmont	2018	Clear, 5 mm	*Autographiviridae*
pphageB51	Soil	Boves (CN)	*Phaseolus vulgaris* cv. *Billò*	Agrion	Piedmont	2018	Clear, 5 mm	*Autographiviridae*
pphageB101	Soil	Boves (CN)	*Phaseolus vulgaris* cv. *Billò*	Agrion	Piedmont	2018	Clear, 5 mm	*Autographiviridae*
pphageT11	Soil	Tetti Pesio (CN)	*Phaseolus vulgaris* cv. *Borlotto nano*	Agrion	Piedmont	2018	Clear, 5 mm	*Autographiviridae*
pphageT12	Soil	Tetti Pesio (CN)	*Phaseolus vulgaris* cv. *Borlotto nano*	Agrion	Piedmont	2019	Clear, 5 mm	*Autographiviridae*
pphageT13	Soil	Tetti Pesio (CN)	*Phaseolus vulgaris* cv. *Borlotto nano*	Agrion	Piedmont	2020	Clear, 5 mm	*Autographiviridae*
pphageT21	Soil	Tetti Pesio (CN)	*Phaseolus vulgaris* cv. *Borlotto nano*	Agrion	Piedmont	2018	Clear, 5 mm	*Autographiviridae*
pphageT22	Soil	Tetti Pesio (CN)	*Phaseolus vulgaris* cv. *Borlotto nano*	Agrion	Piedmont	2018	Clear, 5 mm	*Autographiviridae*
pphageT23	Soil	Tetti Pesio (CN)	*Phaseolus vulgaris* cv. *Borlotto nano*	Agrion	Piedmont	2018	Clear, 5 mm	*Autographiviridae*
pphageBV2	Soil	Verona (VR)	*Phaseolus vulgaris* cv. *Spagnolet (Lamon)*	Phytosanitary Inspection Service	Veneto	2019	Clear, 5 mm	*Autographiviridae*
pphageBV4	Soil	Verona (VR)	*Phaseolus vulgaris* cv. *Spagnolet (Lamon)*	Phytosanitary Inspection Service	Veneto	2019	Clear, 5 mm	*Autographiviridae*
pphageBV71	Soil	Verona (VR)	*Phaseolus vulgaris* cv. *Canalino (Lamon)*	Phytosanitary Inspection Service	Veneto	2019	Clear, 5 mm	*Autographiviridae*
pphageBV72	Soil	Verona (VR)	*Phaseolus vulgaris* cv. *Canalino (Lamon)*	Phytosanitary Inspection Service	Veneto	2019	Clear, 5 mm	*Autographiviridae*

^a^ The letters in brackets indicate the province. CN stands for Cuneo, VC for Vercelli and VR for Verona. ^b^ For Psa strains the reference strain used for this assay was K7#8, and for Pph strains, Cuneo #6_18. ^c^ The classification in the last column of the table was either through high identity with taxonomically classified phages already present in the database for the 15 phages that were sequenced in their whole genome, or for the other 8, identical TEM observed morphology was confirmed with specific primers designed on the sequenced isolates.

**Table 3 viruses-13-02083-t003:** Summary of the characteristic of the genomes of selected phages.

Phage Name	Accession Number	Bacterial Host	Genome Length	GC Content (%)	ORFs	Hypothetical Proteins	Proteins with Predicted Function	ORFans	tRNAs
psageA1	MT740307	Psa K7 #8	98,780 bp	48.79%	176	76	51	49	14
psageB1	MT354569	Psa K7 #8	112,269 bp	56.47%	161	105	40	16	4
pphageB1	MT354570	Pph Cuneo #6_18	41,714 bp	56.63%	52	23	25	4	0
psageK4	MZ348426	Psa K7 #8	98,440 bp	60.44%	179	112	51	16	18
psageB2	MZ348425	Psa K7 #8	50,739 bp	58.51%	77	47	26	4	0

## Data Availability

The full-length sequences of phage genomes were submitted to GenBank and are accessible with the accession numbers MT740307 (psageA1), MT354569 (psageB1), MZ348426 (psageK4), MZ868713 (psageK4e), MZ868718 (psageK9), MZ348425 (psageB2), MT354570 (pphageB1), MZ868714 (pphageB21), MZ868715 (pphageT21), MZ868716 (pphageT12), MZ868717 (pphageBV72). The raw data from the sequencing of 15 phages are available on SRA with accession number PRJNA757206.

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
