# Peer review of "Molecular Characterization and Taxonomic Assignment of Three Phage Isolates from a Collection Infecting Pseudomonas syringae pv. actinidiae and P. syringae pv. phaseolicola from Northern Italy"

_viruses, 2021, doi:10.3390/v13102083_

Round 1
Reviewer 1 Report
The work describes the isolation and characterisation of phages that are infective against Pseudomonas spp. As currently written the work is confusing at times to follow and does not seem to have a logical flow. There is a substantive amount of work done, that has some really interesting insights. However, it is often hidden by what seems unnecessary information, or the context is not explained.
Initially it is stated 20 phages are isolated, then data is presented on the resistance to UV etc and host range on only a subset, without any explanation of why. Then genomic analysis is presented showing phages are similar, which hints as to why a subset are used. But there is no explanation of this. Some explanation is really required and changing the order presented might help the flow of the manuscript
The names of the phages are confusing with phage names in the Genbank files and used in the manuscript not matching (one genome also seems not to be public yet). Given 20 phages were sequenced, all of this raw data should be submitted and the assembled genomes with accession numbers included (table 2) would be a good place for accessions.
The title suggests taxonomic assignment of all phages. This has only been done to varying degrees and needs improvement. It needs to be made clearer what is a new species and new genera, possibly subfamily. Currently the data presented does not support a new subfamily as genera within the current genera within closest families have not been included, thus there is not accurate placement. The term “novel phage” is vague. Cleary some of these phage are new species and genera based on current ICTV standards. Providing suggests names for these in binomial form would be tremendous use to the community moving forward
Minor points
Line 118 0.45 μm pore size membrane. Rather than 0.45 μm membrane
L142 –please clarify if only one spot was used to asses infection or a dilution series ?
L153 – Why were only 3 phages imaged ?
L160 space between units and numbers 1 h . here and elsewhere
L178 – version of SPAdes and settings required. The coverage is also important to know given the SNP calling. Were the genomes corrected post assembly via pilon (or alternative). Parts of the text suggests 600x coverage was used, which is known to introduce errors. Which has subsequent effect on SNP calling
L183- it is not clear how termini was determined. Alignment with the closest genomes only allows this, the closest genome had its termini determined. It is not clear if this is the case for the genomes used.
L189 – version an settings of SNIPPY
L237 italics for Caudovirales and it’s an order not a family.
L252 why were only four phages tested for sensitivity to abiotic parameters , when 20 were isolated ?
Line 255 –I am not sure a greater than 99.9% reduction in titre should be described as stable ? Activity was maintained, but there is a significant drop in titre based on the Figures presented , with 2/3 log drops in titre ?
L279 – state in the methods how this was determined. What virulence database was used to check this
L282 – why are only five phages presented if 20 were isolated ?
L287 – All phages were sequenced then. If they are not identical as stated, the data on the genomes should be made available (even if they are, it should still be made avalible) . The isolation of very similar phages is of interest and implications for this study.
L291 – this seems a curious choice. How can something that only aligns over 3% , be used to orient a genome ?
L292 – it is not clear how phage phageB1 is the same genus as MR5 etc . These phages are 88% identical over ~50% of the sequence. Guidelines for ICTV taxonomy suggest “70% nucleotide identity of the full genome” (https://doi.org/10.3390/v13030506) . These do not meet this criteria when % genome coverage is included – as is required
L294 here it might be better to state it is a new species, rather than a novel genome.
L296- this detail on coverage is good to have and should be included for all phages. But seems out of place in this sentence and no information is provided for other phages. A supplementary table of this information would help
L300 – please provide reference for most phages having organisation or remove statement
L303 – given the termini have not been established, surely the extremity is arbitrary ?
L306 – please provide locus_tag or accession number of the protein in brackets
L307 – again please provide locus_tag for these proteins.
L308 why a NAD/FAN utilising enzyme is involved in cell division and not any other metabolic process is not clear. Please clarify why cell division. No evidence is presented of their origin
L314- here and elsewhere please provide locus_tags or protein accessions when referring to particular proteins/genes in a genome.
L317 – the description of ends is relative. The authors haven’t worked out the temini of the genome
L320 It is not clear what a “host” protein is. PhoH found in phage are very different to the same named protein in bacteria. Why is a protease a host protein ? are the closest relatives actually from the host ?
L326 – there seems to be a jump in logic here. No data is presented to show which phages are the same species. Please submit the raw reads of these other phages.
L328 – I am slightly confused here. Is L326 in regards to blastn ? . And why as a variant analysis not used for these phages like the following phages.
L326 – 345 . These phages are clearly not 100% identical . The genome and raw reads of all phages (including the 100% identical – if they came from different isolates, which the methods suggests. Should ALL be submitted to the ENA and SRA).
L338 – the data as presented does not allow comparison of this. While the analysis allows difference to the common reference , it does not compare difference between isolates
This also raises questions on the logic of the work. As currently written, phages were isolated as subset were selected for testing in different parameters and then all of them sequenced. The logic of this does not seem to make sense. Having the genomic data before the characterisation would explain why so few were characterised, if representatives were being picked. But given the level of SNPs that are different between phages, the phenotypic properties are not likely to be identical
L349 – if these phage are identical, then analysis of one of these is clearly rationalised. However, calling them by the same phage name is not. The methods suggest these phages were isolated independently and are different isolates. They are the same species and strain as each other, but still independent isolates. Furthermore, the naming in the manuscript is different from the naming in the Genbank file , which adds further confusion. Eg psageA1 is named phage phiK7A1 in ENA/Genbank ….
Given the recent adoption of binomial system for phage names, I would suggest using this to differentiate between phage names and taxa.
L350 Again please clarify what a new phage is. Based on the % similarity to its closest relative, it is clearly a new species and genus. Please cite latest ICTV guidelines
L360 it would help to name some of these genes that are referred too. Based on the figure 3C, it is not clear what evidence that these genes are bacterial origin .
L384- here and elsewhere italics for names of phage taxa. Myoviridae etc
L386- Again these phages are not identical and should have separate phage names, but a single species names. All genomic data should be submitted
L390 – It is not clear what the relevance of this data is ? it is clearly not the same species as these other phages and the same species as phiPsa1 . So why mention the others ?
L398- again please refer to locus_tags of proteins in brackets, so readers can find the protein easily in the genbank file
L403 – reference is missing
L405 – similar comment. These phages are not identical so both should be submitted separately. They are a single species, but different isolates. It would also be useful to know the differences between the isolates
L418 pphageB1 has been identified as a member of the Krylovirinae. Minor typo in the spelling on line 422 and the figures. The authors then suggest the phage is a new subfamily based on the clustering outside of the other subfamilies. However, Krylovirinae contains four genera – only 1 is provided in the tree. It is not clear if these phages would fit into the other 3 genera within this subfamily. This analysis needs to be done before claims of new subfamilies. In addition VIRIDIC analysis provides the information to determine if the phage is genus – which has 70% across 100% of the genome. This information can be extracted. Looking at the figure it appears to be the same genus as other phages – but it hard to judge based on shading alone. Clearly marking phages of the Krylovirinae from all genera is also required in Figure 5. Figure 4A does not support a new subfamily , based on the brackets marking subfamilies.
L429/430 . The branch lengths in Figure 4B suggest the psageA1 is more closely relate to other phages in Flaumdravirus and Otagovirus, than Psa267. Why is Psa267 or particular importance ? The methods suggest proteins were chosen based on top blast hits. Yet PsageA1/phiK7A1 is only compared against Pseudomonas phages ? There are multiple hits to phages outside of Pseudomonas phages that his phage hits (based on the genes listed for tree construction). By not including any of these in the tree, it is impossible to get an accurate taxonomic position of the phage.
L447 – what is meant by distinct. Unless the phage are 100% identical then they have only tested host range for a subset of their phages. Which given the application is suggests to be phage therapy, they do not know what the effect of 100s of SNPs has on host range or physical properties. Suggesting phages that are not 100% identical have the same properties is misleading without any evidence. Ideally the host range should have been done on each phage isolate (that was not 100% identical to something else) or the labelling should be of the isolate that it was tested on
L447 – 469 This should be written to reflect only one phage strain of a phage was tested, rather than all of them
L495 – given recent changes this would be myovirus like particles
L513- what is the phik4 isolate ?
psageB2 is not currently publically available. Therefore, this cannot be verified .
L520-527 – this is clearly an interesting point. But at no point are the differences in proteins/genes easily identified in the information provided
L526 – how many genes are different – this information is important to know?
L528 –some further explanation of why alteration in the lysis protein effects the host range, is required.
L529- there is little doubt this is a new species and most likely a new genus as suggested. However the taxonomic position is not clear. Many phages were excluded from the phylogeny that would allow more accurate placement
L555 – given this discussion. It is not clear why only a subset where then used
L558/9 the establishment of a new subfamily is not clear. As mentioned earlier, if these phages are a genus should first be established. There are well established methods for doing this and cutoffs for genera. Methods such as vcontact2 or a more robust phylogenetic analysis with comparison to comprehensive set of phages.
L562 – given the termini of this phage were not established via the sequencing method , how is this known ?
573- it would help to explicitly state the phage is a new species and represents a new genus
Author Response
Please see attached word file with point by point reply

Reviewer 2 Report
Please see the attachment.
Author Response
Please see attached file with point by point reply to your queries

Reviewer 3 Report
This manuscript describes new phages against two pathovars of Pseudomonas syringae. Many phages have already been isolated for pv. actinidiae but a lower variety of pv. phaseolicola phages are known. In the present study, a total of 26 phages were isolated on different strains and a few of them were analysed in details including an analysis of their host range. The genome of some of them was sequenced allowing a description of their genetic characteristics. The title is somewhat inaccurate as molecular characterization and taxonomic assignment were not performed on the whole library. In fact only five phages were investigated thoroughly. The title should be changed to better reflect the results obtained.
There are several points that need to be clarified. The susceptibility of a bacterial strain to a phage cannot be assessed by simply spotting a concentrated suspension of phages. The observed lysis zone could be due to a bacteriocin activity, and if it is the result of phage infection, a quantitative evaluation of the phage virulence should be measured. Efficiency of plating must be performed.
There are some discrepancies concerning phages psage B1 and psageB2 that must be corrected. In Paragraphe 3.4.4 and table 1, psageB2 is indicated as a myovirus (no EM analysis is presented). Upon genome examination this phage is highly similar to phiPsa1, a siphovirus, so psageB2 is likely to be also a siphovirus. The EM examination must be repeated.
Lines 453-455: it is stated that “PsageB2 and psageB1 resulted in some cases in turbid plaques as expected from their likely lysogenic nature as implied by their genome sequencing and annotation”. This characteristic was not described in the genomic analysis of psageB1. Is there an integrase? psageB1 is described as potentially performing a lysogenic cycle but on lines 534 to 536, the opposite is said “we did not observe a temperate behavior in our host range”. It must be clearly stated whether phage psageB1 is a lytic or temperate phage.
Line 510 to 512: the statement is confusing “No genes associated to lysogeny were found, except for psageB2, but we cannot exclude that the other phages might display a temperate behavior in other conditions and in other hosts”. The author should explain how this could be possible.
Specific points
On line 93 LB is Lysogeny broth whereas on line 97 LB medium is probably Luria Broth. This must be corrected
Line 133: 108 instead of 108, 700 µL instead of 700
Line 186: Give the https address of online phageA1
Line 257: “efficiency” used twice
Legend of Figure 3 legend should be corrected for English
Line 349: what is phage phiK7A1?
Author Response
Please see attached file with point by point reply

Round 2
Reviewer 1 Report
The authors have made significant changes to the manuscript, that in my opinion make it easier to follow the good work that they carried out.
Although there are now accessions for all phages listed along with a Project number for SRA reads, there are only two biosamples associated with this project, suggesting the others haven`t being created or made public (I think this is most likely) and thus raw data not available. Many of the accessions are also still not public
Reviewer 2 Report
Please see the attachment.

Reviewer 3 Report
This revised version is improved and the all the comments of this reviewer have been addressed.
However, now that efficiency of plating has been performed there are still some points that remain unclear.
On the side of Table 1, because of the yellow label, it was not possible to read what was associated to the three dark blue colors. In the legend it is said to indicate a greater efficiency of plating as compared to the strain used to isolate the phage. It seems to be very frequent in many phages (as for phage Psage K4e for example), which is rather surprising. How is this explained?
Is there an explanation for the reduced activity of phages against non-pathogenic strains as stated lines 433-435? Could it be that all the pathogenic strains are phylogenetically very close, explaining the broad host-range of some phages?
Minor comments
Line 74 : should be « 22 Psa »
EM of purified phages, line 137. I am surprised that the resuspended pellet after PEG precipitation could be used directly for EM examination. Was the PEG removed using chloroform?
Line 229: should be “revealing”
Line 246: “we chose”
Lines 418-419: pphage T12 is repeated twice
Round 3
Reviewer 2 Report
Dear Authors
There is no need for revision except the order of heading number of the result section. Please modify them in numerical order.
